# Faster Neural Network Training with Data Echoing

## Abstract

In the twilight of Moore's law, GPUs and other specialized hardware accelerators have dramatically sped up neural network training. However, earlier stages of the training pipeline, such as disk I/O and data preprocessing, do not run on accelerators. As accelerators continue to improve, these earlier stages will increasingly become the bottleneck. In this paper, we introduce "data echoing," which reduces the total computation used by earlier pipeline stages and speeds up training whenever computation upstream from accelerators dominates the training time. Data echoing reuses (or "echoes") intermediate outputs from earlier pipeline stages in order to reclaim idle capacity. We investigate the behavior of different data echoing algorithms on various workloads, for various amounts of echoing, and for various batch sizes. We find that in all settings, at least one data echoing algorithm can match the baseline's predictive performance using less upstream computation. We measured a factor of 3.25 decrease in wall-clock time for ResNet-50 on ImageNet when reading training data over a network.

## 1 Introduction

Over the past decade, dramatic increases in neural network training speed have facilitated dramatic improvements in predictive performance by allowing researchers to train bigger models using larger datasets and to explore new ideas more rapidly. As Moore's law ends, general purpose processors are no longer rapidly becoming faster, but specialized hardware continues to drive significant speedups by optimizing for a narrower set of operations. For example, GPUs and TPUs[1] optimize for highly parallelizable matrix operations, which are core components of neural network training algorithms.

However, neural network training requires more than just the operations that run well on accelerators – a training program may need to read and decompress training data, shuffle it, batch it, and even transform or augment it. These steps exercise multiple system components, including CPUs, disks, network bandwidth, and memory bandwidth. It is impractical to design specialized hardware for all these general operations that involve so many different components. Moreover, these operations are not simply executed once at the start of the training program. Since many of today's datasets are too large[2] to fit into an accelerator's memory or even the host machine's main memory, most large-scale neural network training systems stream over the training data, incrementally reading it from disk, pre-processing it in main memory, and copying successive batches of training examples to the accelerator, which runs the training algorithm. Therefore, each training step involves a mixture of operations that do and do not run on accelerators.

There are workloads where the code running on accelerators consumes only a small portion of the overall wall time, and this scenario will only become more common if accelerator improvements continue to outpace improvements in CPUs. In order to speed up training in these cases, we must either (1) make the non-accelerator work faster, or (2) reduce the amount of non-accelerator work required to achieve the desired performance. Option (1) is appealing but requires substantial engineering labor or problem-specific techniques (e.g. Ying et al., 2018; Kumar et al., 2018). Adding more workers might be too expensive. Instead, we focus on option (2) and explore techniques for reducing the total amount of work spent reading and preparing inputs in the training pipeline.

---

[1] https://www.blog.google/products/google-cloud/google-cloud-offer-tpus-machine-learning/

[2] For example, after decoding and standard pre-processing, the ImageNet dataset (Russakovsky et al., 2015) is 700 GB and the Common Crawl dataset is 6.7 TB.

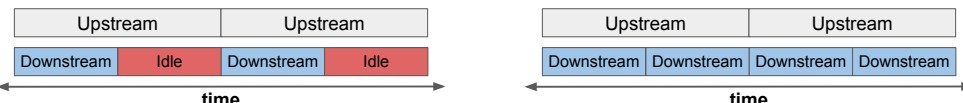

Figure 1: The training pipeline for ResNet-50 on ImageNet, which is representative of many large-scale computer vision programs.

| Upstream | Upstream |
|---|---|
| Downstream | Idle | Downstream | Idle |

**time**

(a) Without data echoing, downstream computational capacity is idle 50% of the time.

| Upstream | Upstream |
|---|---|
| Downstream | Downstream | Downstream | Downstream |

**time**

(b) Data echoing with echoing factor 2 reclaims downstream computational capacity.

Figure 2: The overlapping computation time for pipeline stages upstream and downstream of the data echoing insertion point, if stages are executed in parallel and $t_{\text{upstream}} = 2t_{\text{downstream}}$.

Figure 1 shows the data processing and training pipeline for ResNet-50 (He et al., 2016) on ImageNet, which is representative of many large-scale computer vision programs. First, the training program reads each image from disk, decompresses it into a 3 dimensional array of values, and pushes it into a shuffle buffer. The next stage of the pipeline samples images at random from the shuffle buffer to approximate shuffling the entire dataset, but with a fixed memory budget. The next stage performs pre-processing and data augmentation — each image is randomly cropped and resized to a $224 \times 224 \times 3$ array, then randomly horizontally flipped, and finally has its colors randomly jittered. These random distortions help improve the generalization of vision models, and while these particular operations are specific to images, almost every deep learning pipeline performs some kind of pre-processing on its input data. Finally, images and labels are gathered into batches and sent to the accelerator to perform a step of minibatch stochastic gradient descent (SGD). For brevity, we will refer to the operation that updates the model's parameters for a given batch of training examples as the "SGD step" throughout the paper, even though variants of the basic SGD algorithm are also popular. Our technique applies equally well for any training algorithm that works on successive batches of training examples.

To maximize throughput, the training program is often executed as a pipeline process, so that each stage in Figure 1 operates in parallel from the other stages. Each stage might further employ multiple parallel worker threads or machines. If any of the stages upstream from the SGD step cannot process images at the same rate as the SGD step, the accelerator will be partly idle (see Figure 2a). This can happen for many reasons, including slow transfer from disk or cloud storage, time-consuming pre-processing operations, or inadequate tuning of the number of CPU threads dedicated to each stage of the pipeline. While it can be possible to improve training time by dedicating engineering effort to optimizing the input pipeline, such efforts are often time consuming and can distract from the practitioner's main goal of improving their model's predictive performance. Instead, we propose **data echoing** as a simple, cheap, and effective method for reclaiming idle accelerator capacity. Rather than waiting for more data to become available, we propose simply reusing data that is already available. We do this by adding a stage to the pipeline that repeats (or "echoes") data from the previous stage. Once a practitioner identifies the largest bottleneck in the training pipeline, they can insert an echoing stage after it to reclaim idle accelerator capacity (see Figure 2b).

In this paper, we demonstrate that:

1. data echoing reduces the amount of upstream computation needed to reach a competitive out-of-sample error rate on various datasets and model architectures;
2. data echoing can provide a walltime speedup in practice;
3. data echoing can support a wide range of echoing factors;
4. the effectiveness of data echoing depends on the insertion point in the training pipeline;
5. data echoing can benefit from additional shuffling after echoing, but does not require it; and
6. countering expectations, data echoing reaches the same final error rate as well-tuned baselines.

## 1.1 RELATED WORK

Data echoing shares similarities with experience replay (Mnih et al., 2015), which samples batches from a buffer containing a reinforcement learning agent's past experiences to prevent the most recent interactions from dominating the updates. Although both data echoing and experience replay reuse previous data, our implementation of data echoing specifies the number of times to repeat each example, whereas most implementations of experience replay do not control this explicitly. In addition, local SGD algorithms (Zinkevich et al., 2010; Zhang et al., 2016), which perform multiple local model updates before communicating globally, can also be viewed as reusing data to save computation. However, local SGD targets communication overhead between workers and thus is orthogonal to data echoing.

We are aware of two previous papers that describe variants of data echoing. Fischetti et al. (2018) describe a special case of data echoing they call "minibatch persistency" that reuses minibatches for multiple consecutive SGD updates. They run experiments on CIFAR-10, but do not tune hyperparameters for the baseline or for their method. Neither their method nor their baseline reach competitive test set numbers in their experiments, leaving open the question of whether minibatch persistency has an advantage over a well-tuned baseline. Similarly, Hoffer et al. (2019) describe a special case of data echoing they call "batch augmentation" that repeats examples multiple times within a given batch, but with different augmentations. None of their experiments tune optimization hyperparameters, although their baselines use settings taken from the original papers that introduced each model. Both Fischetti et al. (2018) and Hoffer et al. (2019) primarily motivate their work as methods to improve generalization, only tangentially mentioning the possibility of reclaiming idle computational capacity. We would not expect data echoing to improve generalization for a fixed batch size and number of SGD updates, since then repeated data would be *more* valuable than fresh data. Our experiments in Section 3 only show that data echoing can achieve better out-of-sample error for the same amount of *fresh* data.

Another related line of work accelerates neural network training by subsampling examples non-uniformly (Katharopoulos & Fleuret, 2018). These methods are superficially similar to data echoing, in that their goal is also faster training and that this is accomplished by modifying the examples provided to the accelerator, but fundamentally these approaches work best in different scenarios. With example subsampling, each training example is processed, on average, less than once per epoch, and faster convergence is achieved by reducing the overall time the accelerator spends processing examples. With data echoing, the opposite is true: each training example is processed, on average, more than once per epoch, and faster convergence is achieved by using more accelerator computation per example instead of less. Data echoing does not affect the training data distribution, so it does not change the optimal decision boundary. However, when using non-uniform subsampling, care must be taken to ensure that one converges to the correct minimizer of the non-sampled loss. This is extremely relevant in probability matching tasks like language modeling, but should not be ignored even in other models. In principle, nothing prevents a particular training pipeline from adopting both methods, though systems considerations might make it difficult to observe an actual speedup.

## 2 DATA ECHOING

We implement data echoing by inserting a stage in the training pipeline that repeats (echoes) the outputs of the previous stage. In TensorFlow's (Abadi et al., 2016) `tf.data` library, an echoing stage is as simple as

```
dataset.flat_map(lambda t: tf.data.Dataset.from_tensors(t).repeat(e))
```

where $e$ is the data echoing factor, the number of times each data item is repeated. In some cases, we also shuffle the outputs of the echoing stage, but this can require additional memory. If the overhead of repeating data is negligible and the stages on either side of echoing are executed in parallel (e.g. Chien et al., 2018), then the average time for data echoing to complete one upstream step and $e$ downstream steps is

$$\max \{t_{\text{upstream}}, e \times t_{\text{downstream}}\}, \tag{1}$$

where $t_{\text{upstream}}$ is the time taken by all stages upstream of echoing, $t_{\text{downstream}}$ is the time taken by all stages downstream of echoing, and $e$ is the echoing factor. Non-integral echoing factors can be

achieved in expectation by probabilistically repeating data items. We assume that $t_{\text{upstream}} \geq t_{\text{downstream}}$ throughout the paper, since this is the primary motivation for using data echoing. If we denote the ratio of upstream-to-downstream processing time by $R = t_{\text{upstream}}/t_{\text{downstream}}$, then the time to complete one upstream step and $e$ downstream steps is constant for all echoing factors less than or equal to to $R$. In other words, if $e \leq R$, the additional downstream steps per upstream step are "free" because they utilize idle downstream capacity.

Data echoing aims to decrease training time by reducing the number of upstream steps required to achieve a target predictive performance. When using data echoing, each upstream step is used for $e$ (instead of 1) downstream SGD updates. If the required number of SGD updates with data echoing is the same as without, then training time will decrease by a factor of $e$. However, since repeated data might be less valuable than completely fresh data, data echoing might require more downstream SGD updates to reach the desired predictive performance, and so the speedup factor might be less than $e$. We investigate the effect of data echoing on training time in Section 3.

Given that every operation in the training pipeline takes some time to execute, the amount of idle downstream time that data echoing can exploit is greatest if the echoing stage is applied just before the SGD update. For the ResNet-50 training pipeline in Figure 1, this would result in the same minibatch of training examples being used multiple times per epoch. However, we might prefer to insert data echoing earlier in the pipeline if it provides a more favorable trade-off between the number of upstream steps and downstream steps. In particular, the following factors influence the behavior of data echoing at different insertion points:

**Echoing before or after batching:** Echoing before batching means data is repeated and shuffled at the example level instead of the batch level. This increases the likelihood that nearby batches will be different, at the expense of potentially duplicating examples within a batch. Whether diversification across batches or within batches is more important is an empirical question that we address in Section 3. We call the class of algorithms that echo before batching *example echoing* and the class of algorithms that echo after batching *batch echoing.*

**Echoing before or after augmentation:** Echoing before data augmentation allows repeated data to be transformed differently, potentially making repeated data more akin to fresh data. Methods like dropout that add noise during the SGD update can similarly make repeated data appear different (Hoffer et al., 2019), even in the absence of augmentation or when echoing after augmentation.

The behavior of data echoing is also influenced by the amount of shuffling (if any) performed after the echoing stage. Adding a shuffle buffer increases the likelihood that nearby SGD updates use different data, which is likely beneficial. The larger the buffer size, the more repeated data are shuffled, and the closer the training procedure approximates a program that loads the entire training set in memory before sampling data at random. However, since we are focused on large-scale workloads, we assume that we can only afford a buffer size that is a relatively small fraction of the (augmented) dataset size.

## 3    EXPERIMENTS

We evaluated data echoing on two language modeling tasks, two image classification tasks, and one object detection task. For language modeling, we trained the Transformer model (Vaswani et al., 2017) on the LM1B (Chelba et al., 2014) and Common Crawl[3] datasets. For image classification, we trained ResNet-32 (He et al., 2016) on the CIFAR-10 dataset (Krizhevsky & Hinton, 2009), and ResNet-50 on the ImageNet dataset (Russakovsky et al., 2015). For object detection, we trained the Single Shot Detector (SSD, Liu et al., 2016) on the COCO dataset (Lin et al., 2014).

The primary question we investigated was whether data echoing could provide a training speedup. We measured training cost as the number of "fresh" training examples[4] required to reach a target out-of-sample metric value. The number of fresh examples is proportional to the number of upstream steps in the training pipeline, and therefore proportional to wall time if the echoing factor is less than or equal to the ratio of upstream-to-downstream processing time, $R$ (see Section 2). We did not assume or measure the value of $R$ in most of our experiments, since $R$ depends on the implementation and sometimes on irregular factors like network congestion. Not all of our tasks satisfied $R \geq 1$ in

---

[3]http://commoncrawl.org/2017/07/june-2017-crawl-archive-now-available/
[4]Each time a training example is read from disk, it counts as a fresh example.

Table 1: Tasks summary.

| Model | Dataset(s) | Task | Evaluation metric | Target |
|---|---|---|---|---|
| Transformer | LM1B, Common Crawl | Language modeling | Cross entropy | 3.9 |
| ResNet-32 | CIFAR-10 | Image classification | Accuracy | 91% |
| ResNet-50 | ImageNet | Image classification | Accuracy | 75% |
| SSD | COCO | Object detection | mAP | 0.24 |

all experiments. Instead, we designed most of our experiments to investigate whether data echoing could reduce the number of fresh examples needed across various tasks, since this measurement is implementation independent. We confirm that the number of fresh examples is a proxy for walltime in Section 3.2.

For each workload, we ran an initial set of experiments without data echoing and tuned the hyperparameters to achieve the best out-of-sample performance within a practical computational budget.[5] We selected the target metric value to be slightly worse than the best observed in the initial experiments to ensure it could be reached reliably. We verified that small changes to our targets did not affect our conclusions. Table 1 summarizes the workloads and target metric values we used in our experiments.

We trained the SSD model using SGD with momentum (Polyak, 1964; Rumelhart et al., 1986) and the Transformer and ResNet models using Nesterov momentum (Nesterov, 1983; Sutskever et al., 2013). We used a constant learning rate for Transformer, and we used learning rate schedules for ResNet (linear decay) and SSD (linear warmup for 3.5 epochs followed by piecewise exponential decay). We preprocessed the text datasets identically to Shallue et al. (2018). We augmented the image datasets at training time by resizing each image, taking a random crop, and randomly horizontally reflecting the cropped images. We randomly distorted the image colors for ImageNet and COCO. Unless otherwise specified, we used a batch size of 1024 for Transformer and ResNet-50, 128 for ResNet-32, and 256 for SSD. We used batch normalization (Ioffe & Szegedy, 2015) for ResNet-50 and SSD with virtual batch sizes (Hoffer et al., 2017) of 32 and 128, respectively.

In each experiment, we independently tuned the learning rate, momentum, and, where applicable, the parameters governing the learning rate schedule[6]. We manually chose the search spaces based on our initial experiments, and we verified after each experiment that the optimal hyperparameter values were away from the search space boundaries. We used quasi-random search (Bousquet et al., 2017) to tune the hyperparameters with fixed budgets of non-divergent[7] trials (100 for Transformer and ResNet-32, and 50 for the more expensive ResNet-50 and SSD models). We then chose the trial that reached the target metric value using the fewest number of fresh examples. We repeated this hyperparameter search 5 times for each search space. All figures in this section show the mean number of fresh examples required over these 5 searches, with the minimum and maximum shown as error bars. Appendix A shows the average training curve over these 5 searches for every experiment in this section.

The baseline training program for all of our workloads is shown in Figure 1. Our experiments evaluated the effects of adding data echoing to various points in the training pipeline. We considered three variants of data echoing: example echoing before augmentation, example echoing after augmentation, and batch echoing. For the example echoing variants, we omitted the baseline's "shuffle examples" buffer and inserted a shuffle buffer after the echoing stage with the same size as the baseline's buffer. For batch echoing, we kept the baseline's shuffle buffer and repeated batches without shuffling after the "batch examples" stage. Therefore, our training pipeline always had one shuffle buffer with the same size in all cases, so all data echoing variants used the same amount of memory as the baseline.

---

[5] 20k steps for LM1B, 60k for Common Crawl, 110k for ImageNet, 150k for CIFAR-10, and 30k for COCO.

[6] For ResNet, we tuned the number of steps over which the learning rate would linearly decay by a factor of 100 (for CIFAR-10) or 1,000 (for ImageNet). For SSD, we tuned the two epoch numbers at which the learning rate decayed by a factor of 10.

[7] We discarded trials that diverged, which typically occurred when the learning rate was too high.

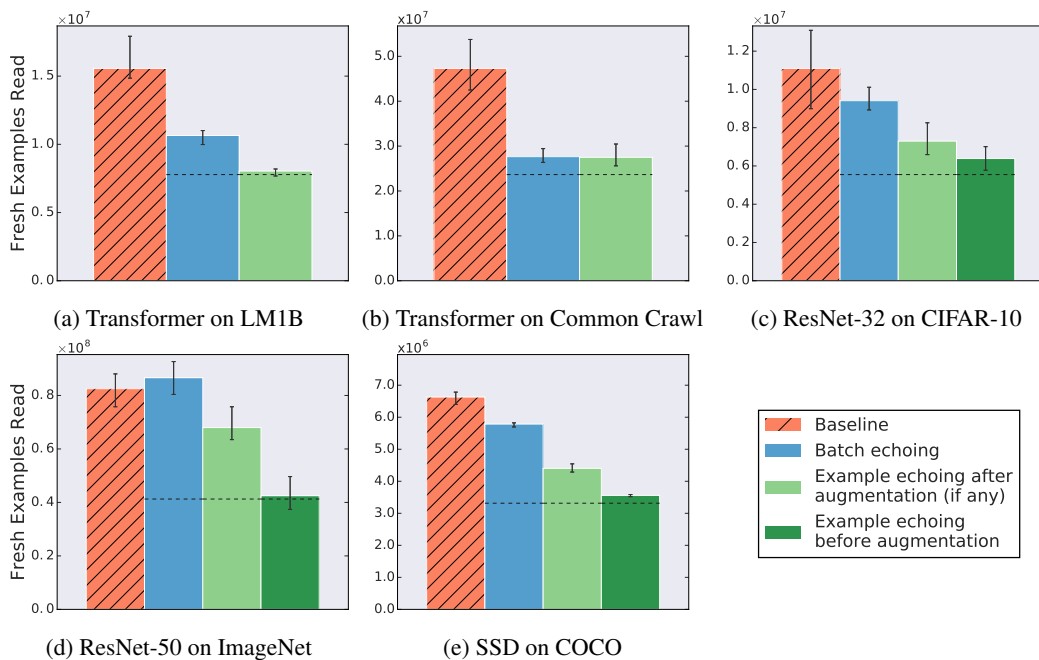

Figure 3: Data echoing with echoing factor 2 either reduces or does not change the number of fresh examples needed to reach the target out-of-sample performance. Dashed lines indicate the expected values if repeated examples were as useful as fresh examples.

We used buffer sizes of $10^6$ for LM1B and Common Crawl, $10^4$ for CIFAR-10, $10^5$ for ImageNet, and $10^4$ for COCO. We explored the effects of increasing the buffer sizes in Section 3.5.

### 3.1 DATA ECHOING CAN REDUCE THE NUMBER OF FRESH EXAMPLES REQUIRED FOR TRAINING

Figure 3 shows the effect of data echoing with echoing factor 2 for all workloads in Table 1. In all but one case, data echoing requires strictly fewer fresh examples than the baseline to reach the target out-of-sample performance. The sole exception (batch echoing on ResNet-50) requires about the same number of fresh examples as the baseline – data echoing provides no benefit, but does not harm training either. The earlier echoing is inserted in the pipeline, the fewer fresh examples are needed: example echoing requires fewer fresh examples than batch echoing, and echoing before data augmentation requires fewer fresh examples than echoing after. We did not observe any negative interaction between data echoing and batch normalization for ResNet-50 or SSD.

### 3.2 DATA ECHOING CAN REDUCE TRAINING TIME

If the training time is dominated by upstream operations like reading and pre-processing input data, data echoing should provide a walltime speedup proportional to the reduction in the number of fresh examples needed for training. To confirm this, we constructed a training pipeline dominated by input latency by streaming our training data from a cloud storage service. While this setup was somewhat contrived, streaming training data over a network is realistic for many large-scale production workloads. For ResNet-50 on ImageNet, the ratio of upstream-to-downstream time in this training pipeline was approximately $R \approx 6$, although the exact number fluctuated with the network transfer rate.

We ran experiments with ResNet-50 on ImageNet with echoing factors $e$ between 1 and 5. Since the bottleneck was transferring data from cloud storage, we inserted data echoing early in the training pipeline before data augmentation. We tuned the hyperparameters separately for each echoing factor, and selected the hyperparameters that reached 75.3% validation accuracy the fastest. We then ran the optimal hyperparameters 5 times for each echoing factor and recorded the number of fresh

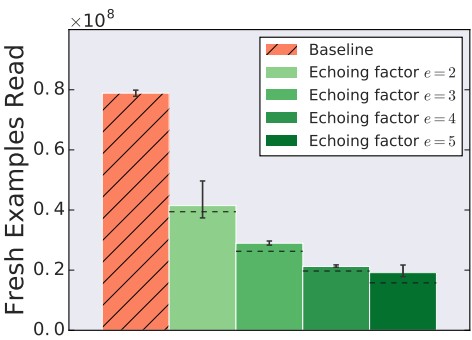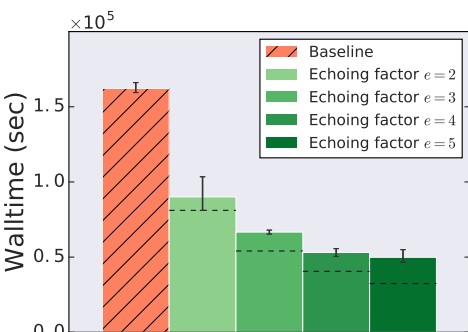

Figure 4: Example echoing before augmentation can reduce training time for ResNet-50 on ImageNet. Dashed lines indicate the expected values if repeated examples were as useful as fresh examples and there was no overhead from echoing.

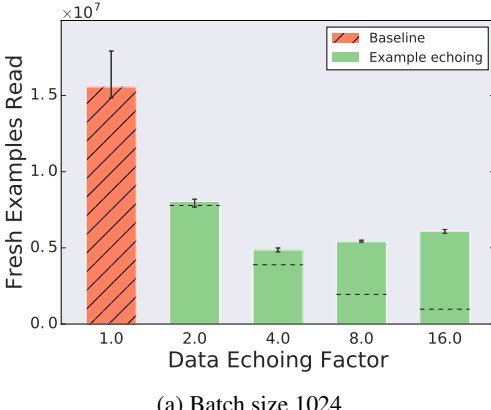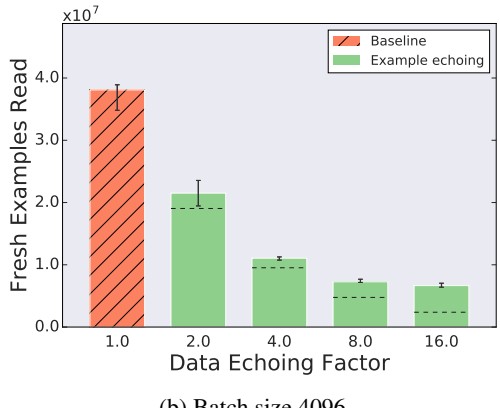

(a) Batch size 1024

(b) Batch size 4096

Figure 5: Example echoing reduces the number of fresh examples needed for Transformer on LM1B for echoing factors up to (at least) 16. Dashed lines indicate the expected values if repeated examples were as useful as fresh examples.

examples and walltime required to reach 75% accuracy. Figure 4 shows that data echoing does indeed provide a walltime speedup proportional to the number of fresh examples read over the network. The reduction in walltime is slightly lower than the fractional reduction in fresh examples because our implementation of data echoing incurs a slight overhead. Nonetheless, data echoing provides a significant speedup for all echoing factors, up to a speedup factor of 3.25 for echoing factor 5.

## 3.3 DATA ECHOING CAN BE USEFUL UP TO A REASONABLE UPPER BOUND ON THE ECHOING FACTOR

Figure 5 shows the effect of example echoing with echoing factors up to 16 for Transformer on LM1B. For batch size 1024, the maximum useful echoing factor is somewhere between 4 and 8; beyond this value, the number of fresh examples required is larger than for smaller echoing factors. As the echoing factor increases, the number of fresh examples required must eventually exceed the baseline, but even an echoing factor as large as 16 still requires significantly fewer fresh examples than the baseline. For batch size 4096, the maximum useful echoing factor is even larger than 16, suggesting that larger batch sizes can support larger echoing factors than smaller batch sizes.

## 3.4 DATA ECHOING AS BATCH SIZE INCREASES

With larger batch sizes, batch echoing performs better, but example echoing sometimes requires more shuffling. Figure 6 shows the effect of data echoing with echoing factor 2 for different batch sizes. As the batch size increases, the performance of batch echoing relative to the baseline either stays the

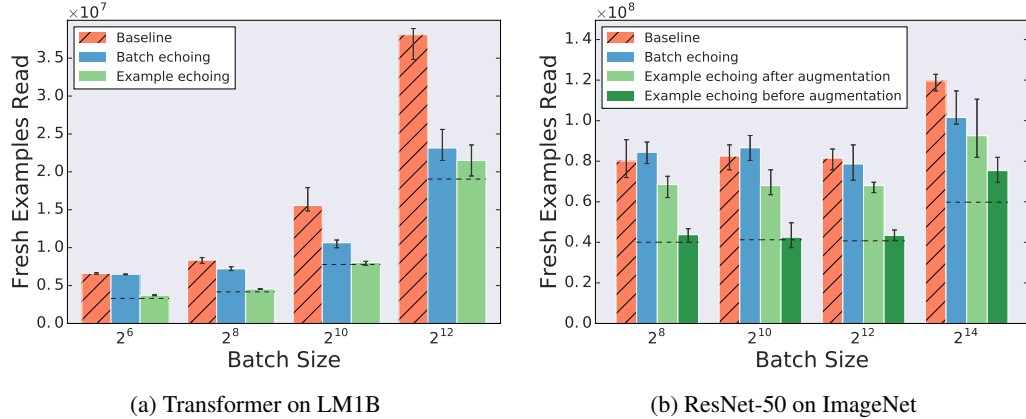

(a) Transformer on LM1B                    (b) ResNet-50 on ImageNet

Figure 6: As the batch size increases, the performance of batch echoing relative to the baseline either stays the same or improves, while for example echoing it either stays the same or gets worse. Dashed lines indicate the expected values if repeated examples were as useful as fresh examples.

same or improves. This effect makes sense given that repeated batches should approximate fresh batches as the batch size approaches the training set size, and so, in the limit, batch echoing must reduce the required number of fresh examples by the echoing factor. On the other hand, Figure 6 shows that the performance of example echoing relative to the baseline either stays the same or gets worse as the batch size increases. Since the expected fraction of duplicate examples within each batch increases with the batch size, example echoing with larger batches may behave more like a smaller batch size in practice. A smaller batch size may increase the required number of SGD updates (Shallue et al., 2018), which could explain the example echoing results in Figure 6. Increasing the amount of shuffling for repeated examples (at the cost of additional memory) could improve the performance of example echoing at larger batch sizes by reducing the probability of duplicate examples in each batch.

### 3.5 DATA ECHOING PERFORMS BETTER WITH MORE SHUFFLING

Figure 7 shows the effect of increasing the shuffle buffer size (at the cost of additional memory) for data echoing with echoing factor 2. While all batch echoing experiments in the previous sections repeated batches without shuffling, the performance of batch echoing improves if repeated batches are shuffled, with more shuffling giving increasingly better performance. Similarly, the performance of example echoing improves with increasing shuffle buffer size, even though it does not help the baseline. This is because more shuffling reduces the probability of duplicate examples within each batch, as discussed in Section 3.4.

### 3.6 DATA ECHOING DOES NOT HARM PREDICTIVE PERFORMANCE

Although one might be concerned that reusing data could harm final predictive performance, we did not observe any case where data echoing with a reasonable echoing factor failed to reach our target metric value. To further demonstrate that data echoing does not degrade solution quality, we ran experiments with Transformer on LM1B and ResNet-50 on ImageNet to find the best achievable performance within a fixed budget of fresh examples, both with and without data echoing. We picked the fresh-examples budgets so that the baseline models would achieve at least our target metric values from Table 1. We used an echoing factor of 4 for all data echoing experiments. We tuned the hyperparameters for the baseline and for all data echoing variants using 500 trials for Transformer and 100 trials for ResNet-50. Figure 8 shows the trials that reached the best out-of-sample performance at any point during training for each experiment. All data echoing variants achieved at least the same performance as the baseline for both tasks.

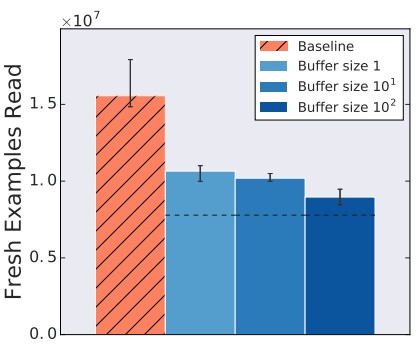 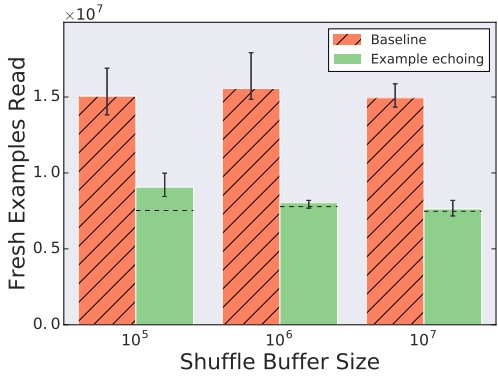

(a) Transformer on LM1B with batch echoing    (b) Transformer on LM1B with example echoing

Figure 7: Data echoing performs better with more shuffling. Dashed lines indicate the expected values if repeated examples were as useful as fresh examples.

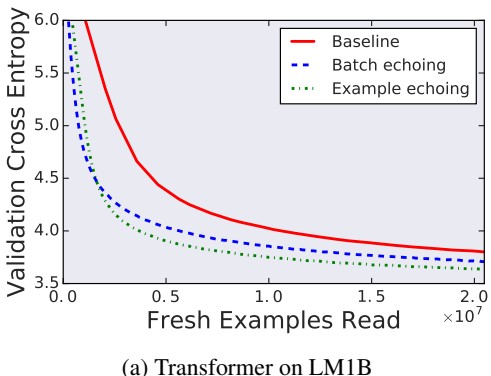 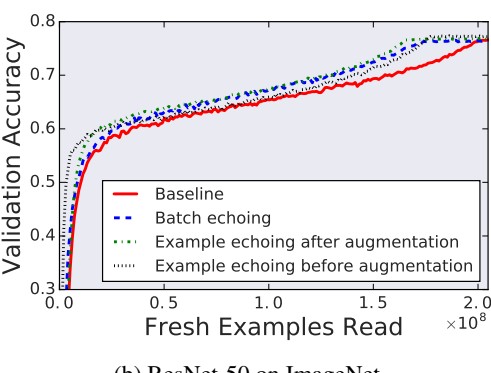

(a) Transformer on LM1B    (b) ResNet-50 on ImageNet

Figure 8: Individual trials that achieved the best out-of-sample performance during training.

## 4 CONCLUSION

Data echoing is a simple strategy for increasing hardware utilization when the training pipeline has a bottleneck in one of the upstream stages. Although *a priori* one might worry that SGD updates with repeated data would be useless or even harmful, for every workload we considered, at least one variant of data echoing reduced the total number of examples we needed to read from disk. This was true even for Transformer on Common Crawl, a dataset so large that we do not even train for a full epoch. In this case, data echoing reached the target predictive performance while seeing only a subset of the examples seen by the baseline. Echoing after augmentation was still effective at reducing the total number of examples read from disk, making it appealing for image datasets that employ expensive data augmentation that runs on the CPU. If reading input data is the bottleneck, then echoing before augmentation will typically provide the greatest speedup. We measured a factor of 3.25 decrease in walltime for ResNet-50 on ImageNet when reading training data over a network.

Data echoing is an effective alternative to optimizing the training pipeline or adding additional workers to perform upstream data processing, which may not always be possible or desirable. Although the exact speedup depends on the model architecture, dataset, batch size, and how well repeated data are shuffled, setting the echoing factor to the ratio of upstream-to-downstream processing time maximizes the potential speedup and worked well in our experiments, even for large ratios. As improvements in specialized accelerators like GPUs and TPUs continue to outpace general purpose computation, we expect data echoing and similar strategies to become increasingly important parts of the neural network training toolkit.

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

# A  BEST TRAINING CURVES FOR ALL EXPERIMENTS

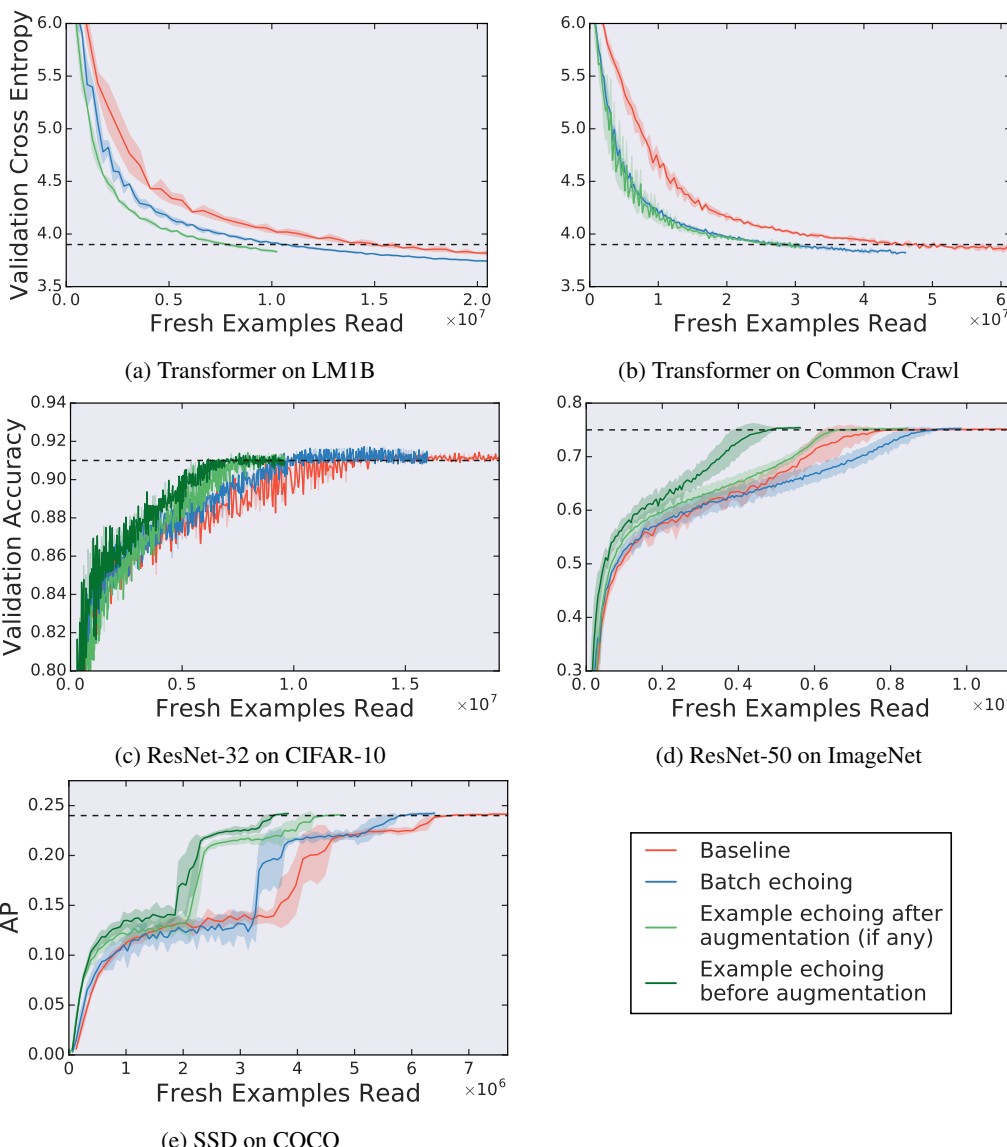

Figure 9: The best training curves for experiments in Section 3.1. Solid lines represent the mean over the best trial from 5 independent hyperparameter searches, and shaded regions represent one standard deviation. The dashed line represents the target metric value.

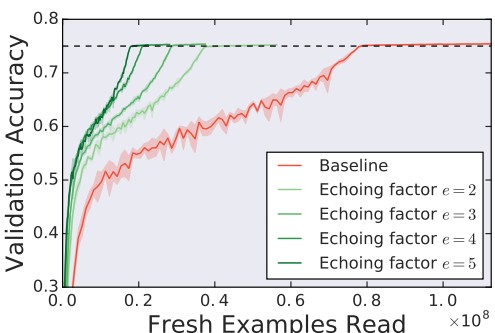

Figure 10: The best training curves for experiments in Section 3.2. Solid lines represent the mean over the best trial from 5 independent hyperparameter searches, and shaded regions represent one standard deviation. The dashed line represents the target metric value.

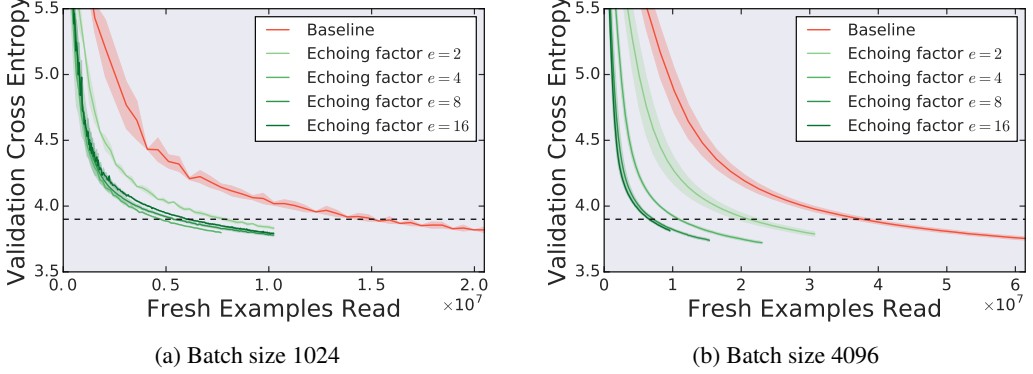

(a) Batch size 1024

(b) Batch size 4096

Figure 11: The best training curves for experiments in Section 3.3. Solid lines represent the mean over the best trial from 5 independent hyperparameter searches, and shaded regions represent one standard deviation. The dashed line represents the target metric value.

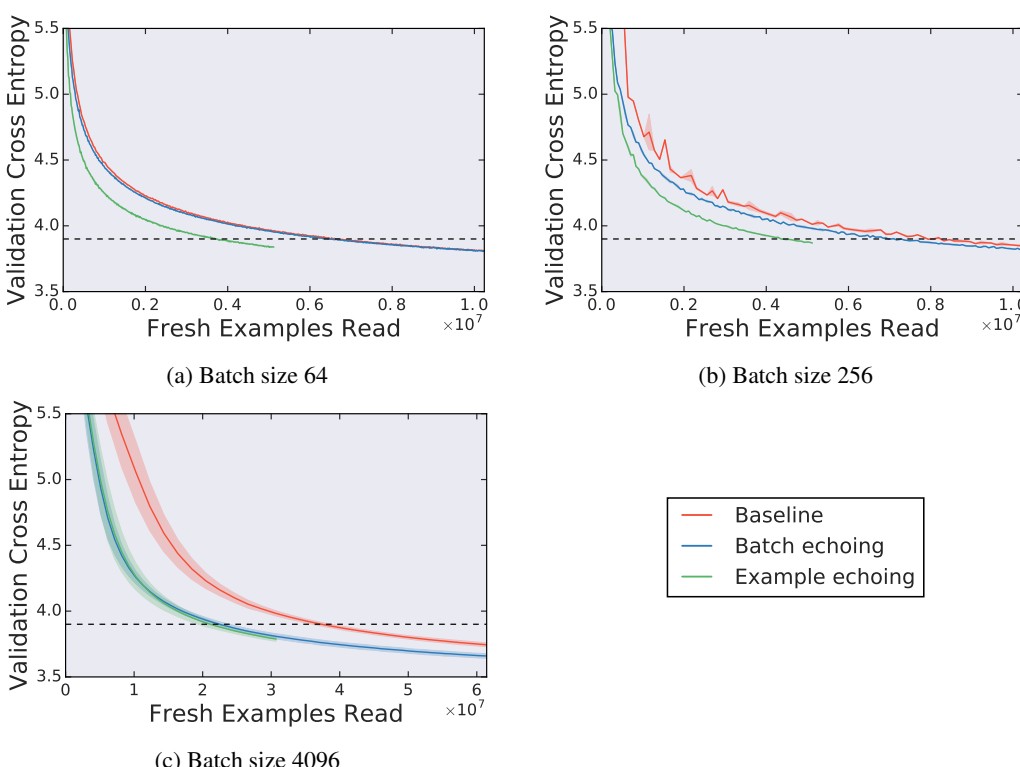

(a) Batch size 64

(b) Batch size 256

(c) Batch size 4096

Figure 12: The best training curves for experiments with Transformer on LM1B in Section 3.4. Solid lines represent the mean over the best trial from 5 independent hyperparameter searches, and shaded regions represent one standard deviation. The dashed line represents the target metric value.

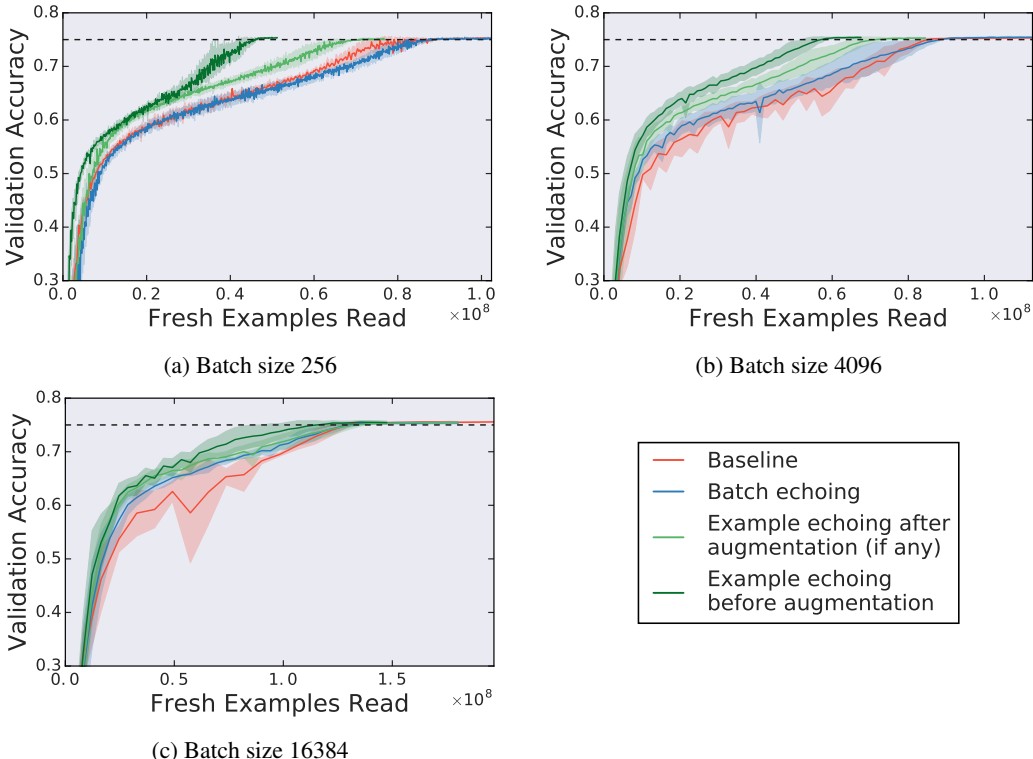

(a) Batch size 256

(b) Batch size 4096

(c) Batch size 16384

Figure 13: The best training curves for experiments with ResNet-50 on ImageNet in Section 3.4. Solid lines represent the mean over the best trial from 5 independent hyperparameter searches, and shaded regions represent one standard deviation. The dashed line represents the target metric value.

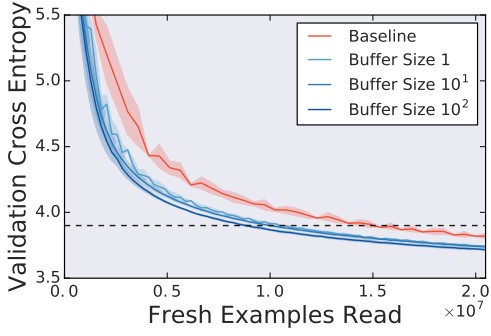

Figure 14: The best training curves for experiments with batch echoing in Section 3.5. Solid lines represent the mean over the best trial from 5 independent hyperparameter searches, and shaded regions represent one standard deviation. The dashed line represents the target metric value.

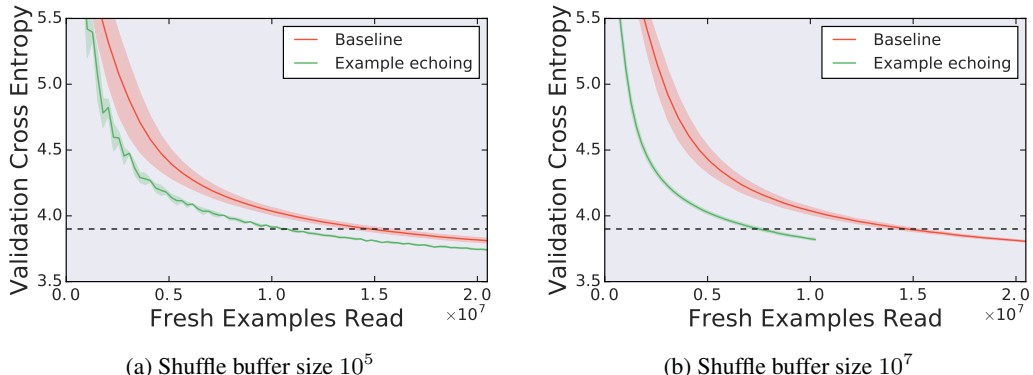

(a) Shuffle buffer size $10^5$

(b) Shuffle buffer size $10^7$

Figure 15: The best training curves for experiments with example echoing in Section 3.5. Solid lines represent the mean over the best trial from 5 independent hyperparameter searches, and shaded regions represent one standard deviation. The dashed line represents the target metric value.

