# OpenReview forum: "Faster Neural Network Training with Data Echoing"
_ICLR.cc/2020/Conference — Reject_

### Official Review · AnonReviewer1 · 2019-10-19
**Official Blind Review #1**

**Rating:** 6

**Review:**

This paper discusses the use of data echoing (re-passing data fetched from drive or cloud) to maximize GPU usage and reduce reliance on data transportation time. The schemes basically are: reusing data at the example level, after data augmentation, or after batching. The experiments measure how much fresh data is needed in order to reach the same level of validation accuracy, with significant speedup when echoing is used.

I thought the paper is very nicely motivated, although this is out of my area so I cannot comment on how thoroughly the problem of data fetching is investigated in other works. The evaluations are also nice, and appropriately uses a large model (Resnet 50) and dataset (Imagenet).

The simplicity of the method is a plus, but I question a fundamental part, especially if batch echoing is used--isn't this just the same as running SGD twice, and therefore halving the stepsize and doubling the number of steps? From the optimization viewpoint, it seems that if less data was used and a good validation error level is reached, then how do we not know that less data wouldn't work well in the first place? I understand that all step sizes and decay rates were chosen independently per experiment; can those numbers be shared in a way to see if this is happening or not? Figure 8 also suggests that though it may take a long time for the baseline to reach the same level as that with batch echoing, everyone reaches a pretty low error rate at about the same point, and the difference may be in the "slow converging" phase of the optimization; thus measuring how long it takes to reach a specific low error rate may be an exaggerated measure.

Basically, what I am saying is that the idea is nice, but the results look a bit magical. I'm happy to increase my score if the authors can upload more intermediary results, like plots of form figure 8, decay rates and schedules, batch sizes, exact repetition schedules, etc.

minor: page 3 end of paragraph 2: "repeated data would NOT be more valuable than fresh data?"

After rebuttal: The authors have addressed my concerns and I more-or-less believe the results. I see why the contribution can be viewed as minor, but it is well-motivated and looks like a nice set of experiments. I encourage the authors to make their code available so that it can be easily incorporated in applications.

**Experience Assessment:**

I do not know much about this area.

**Review Assessment: Checking Correctness Of Derivations And Theory:**

I assessed the sensibility of the derivations and theory.

**Review Assessment: Checking Correctness Of Experiments:**

I assessed the sensibility of the experiments.

**Review Assessment: Thoroughness In Paper Reading:**

I read the paper at least twice and used my best judgement in assessing the paper.

---

> ### Author Response · Authors · 2019-11-10
> **We have incorporated the reviewer’s feedback**
>
> Thanks for the thoughtful comments. We’d like to address some points of confusion:
> ● As the reviewer pointed out, batch echoing with no shuffling is equivalent to just doing two SGD steps on the same data. However, neural networks are not linear functions of the parameters, so two consecutive SGD updates are not equivalent to one SGD update with twice the learning rate, as the gradient is different after doing one SGD update. Note that if they were equivalent, then our highly tuned baselines would outperform the data echoing experiments, which is not what we observe.
> ● We find that most DL models spend a lot of time in the “slow converging” phase of training, and yet this phase is necessary to achieve state-of-the-art metrics on important problems, which is why we measure steps-to-target-metric. The steps-to-target-metric improvement can then be combined with how input-bound a model is to estimate time-to-target-metric improvement, which is what a practitioner would care about.
> ● In page 3 end of paragraph 2, we mean that data echoing can only improve final performance over a baseline with the same batch size and *training step* budget if repeated data are more valuable than fresh data (since the data echoing model would see less fresh data within the training step budget).
>
> We have added plots like Figure 8 for all experiments to Appendix A, and added details about the learning rate schedules we used for all workloads in Section 3. To avoid overwhelming the reader, we did not include all of the 355 best hyperparameter settings (5 hyperparameter searches for each of 71 experiments), but if the reviewer thinks this information is important, we would be happy to include it in the next revision. We hope that we have addressed the concerns and feedbacks of the reviewer, and hope that the reviewer will consider raising their score.

---

### Official Review · AnonReviewer3 · 2019-10-22
**Official Blind Review #3**

**Rating:** 3

**Review:**

The authors propose a simple method for avoiding bottlenecks during NN training, whereby training examples are utilized multiple times per read. The work focuses on cases where the cost of preparing a minibatch exceeds that of a training step (both a forward pass and, subsequently, a parameter update). Working within said regime, the authors investigate different strategies for 'echoing' examples.


Feedback:
  The proposed method itself is very simple: that's fine. While some cursory analysis of data echoing's theoretical implications would be appreciated, I am fine with practically motivated solutions that address real issues. Simple 'tricks' that are easy to implement and widely applicable are often useful tools. Especially when little theoretical analysis is provided, introducing such a trick requires strong empirical evidence to validate its efficacy. As it stands, failure to provide crucial information forces the reader to suspend disbelief when evaluating the proposed method's impact.

  The authors seem to tiptoe around the issue of the relative cost of prefetching a batch versus that of a combined forward pass and parameter update. The work is predicated upon the assumption that the ratio of said costs $R > 1$, but the authors state that an unspecified subset of their experiments violate this assumption. What's more, real-world values of $R$ are not reported (or even measured!). As per Amdahl's law, $R$ upper bounds the potential benefits for data echoing; hence, failing to report $R$ is more than a little bit concerning. By the same token, the appropriate statistic for various result figures would seemingly be time rather than, e.g., the number of fresh examples read. As a reviewer, I would rather see evidence that data echoing provides modest benefits in realistic scenarios than x3.25 speedup in self-described "contrived" examples. Alternatively, consider providing real-world examples where $R > 1$ to help ground your arguments.


Questions:
  - Why was extensive hyperparameter tuning necessary?
  - Why are most results reported in terms of time/steps to achieve a target value? If nothing else, consider providing the corresponding learning curves (as in Figure 8) as an appendix (incl. means and standard errors).


Nitpicks, Spelling, & Grammar:
  - Metaparameters -> hyperparameters
  - Streamline list at end of introduction:
    """
    In this paper, we demonstrate that data echoing:
        1. reduces the...
    """

**Experience Assessment:**

I do not know much about this area.

**Review Assessment: Checking Correctness Of Derivations And Theory:**

N/A

**Review Assessment: Checking Correctness Of Experiments:**

I assessed the sensibility of the experiments.

**Review Assessment: Thoroughness In Paper Reading:**

I read the paper at least twice and used my best judgement in assessing the paper.

---

> ### Author Response · Authors · 2019-11-10
> **Input-bound workloads are a moving target, but in our experience most large-scale models are initially input bound**
>
> Thank you for the thoughtful comments and spelling/naming suggestions. We have updated our draft to use “hyperparameter” instead of “metaparameter” and added additional plots like Figure 8 to Appendix A.
>
> Regarding simple tricks with little theory requiring extensive experimental evaluation, we strongly agree, which is why we trained thousands of models in this paper and very thoroughly tuned the hyperparameters of all our baselines not just for accuracy but for speed of convergence, to convince ourselves both that the speedups are real and that there is no quality loss due to data reuse. We strongly believe that the conclusions in this paper would not be valid without extensive tuning of our baselines and thorough experiments.
>
> Regarding a realistic example, we are very conflicted. In our experience with large-scale commercial applications of deep learning, most large-scale models tend to be input-bound when they are first developed, as reading data from distributed systems and preprocessing it on the fly tends to require a lot of ad-hoc CPU computation. Then, as the models prove themselves useful, a lot of software engineering effort is spent speeding them up (see for example the complexity of the tf.data package, which exists solely to accelerate data preprocessing in TensorFlow, as an example of spending engineering effort optimizing input pipelines). This, however, means we’re working with a moving target, and examples of slowness become “contrived” as software engineering workarounds and optimizations are applied to get the upstream performance to an acceptable range for most well-known models. When a new generation of hardware arrives, this effort must be repeated, as in general CPUs are not getting faster at the same rate as accelerators. So for these reasons we felt it would be hard to point to a single example and say “this one model here is input-bound” since for any example it might be possible to make it not input-bound with enough work and enough resources to parallelize preprocessing. We instead want to propose that this engineering effort is best spent elsewhere and simpler techniques like data echoing can achieve full hardware utilization and faster training for far less engineering effort.
>
> Since we’re evaluating the general case (how much hardware idleness can data echoing possibly eat up) rather than optimizing individual workloads, we focused our experimental results on steps-to-target as this metric is robust to changes in the software or hardware systems involved.

---

### Official Review · AnonReviewer2 · 2019-10-23
**Official Blind Review #2**

**Rating:** 3

**Review:**

This paper proposes a method to compensate the high latency brought by data IO/processing in neural network training. Specifically the authors propose to repeat training on the same subset of data during waiting time for the new data.  In this way, the data efficiency is improved, as verified by thorough experiments on various of real-world tasks.

Although the experiments look promising, I have to say the innovation of this paper is limited. The way of reusing the current data during waiting looks more like a straightforward trick, rather than a novel idea that deserved to be published at ICLR.  Furthermore, I’m wondering that how general the scenarios of “t_{downstream}>t_{upstream}” will be. Even in industrial level applications (e.g., billon level recommendation or click prediction task), AFAIK the training (including feedforward/backprop/communication in the distributed setting) consumes most of the time, while the data reading/preprocessing is comparatively cheap. Last but not least, what will the final performance will be given the potentially harmful consecutive reuse of data? Will it be worse than baseline?


**Experience Assessment:**

I have published one or two papers in this area.

**Review Assessment: Checking Correctness Of Derivations And Theory:**

I assessed the sensibility of the derivations and theory.

**Review Assessment: Checking Correctness Of Experiments:**

I assessed the sensibility of the experiments.

**Review Assessment: Thoroughness In Paper Reading:**

I read the paper at least twice and used my best judgement in assessing the paper.

---

> ### Author Response · Authors · 2019-11-10
> **On the generality of our technique**
>
> Thank you for the comments. Regarding the generality of “t_upstream > t_downstream,” in our industry experience this seems to almost always be the case. And as we mentioned in the abstract and introduction, as GPUs and other accelerator hardware become faster and CPUs do not, it’s highly likely that fairly soon even standard dataset / model combinations will suffer from this problem (let alone applications which need to deliberately train a small model which is cheap to train, like MobileNet, because it will be deployed on a compute-constrained device).
>
> We’d also like to point the reviewer towards Section 3.6 regarding the question about the “potentially harmful consecutive reuse of data”. Note that in all our experimental sections we did not find a noticeable performance loss (versus a highly tuned baseline) from reusing data, which is a primary contribution of this paper. If you think this can be made clearer, please let us know. Otherwise, we hope the reviewer will consider raising their score if the current version of the manuscript addresses their concern.

---

### Decision · Program_Chairs · 2019-12-19

**Decision:**

Reject

**Comment:**

This paper presents a simple trick of taking multiple SGD steps on the same data to improve distributed processing of data and reclaim idle capacity. The underlying ideas seems interesting enough, but the reviewers had several concerns.

1. The method is a simple trick (R2). I don't think this is a good reason to reject the paper, as R3 also noted, so I think this is fine.
2. There are not clear application cases (R3). The authors have given a reasonable response to this, in indicating that this method is likely more useful for prototyping than for well-developed applications. This makes sense to me, but both R3 and I felt that this was insufficiently discussed in the paper, despite seeming quite important to arguing the main point.
3. The results look magical, or too good to be true without additional analysis (R1 and R3). This concerns me the most, and I'm not sure that this point has been addressed by the rebuttal. In addition, it seems that extensive hyperparameter tuning has been performed, which also somewhat goes against the idea that "this is good for prototyping". If it's good for prototyping, then ideally it should be a method where hyperparameter tuning is not very necessary.
4. The connections with theoretical understanding of SGD are not well elucidated (R1). I also agree this is a problem, but perhaps not a fatal one -- very often simple heuristics prove effective, and then are analyzed later in follow-up papers.

Honestly, this paper is somewhat borderline, but given the large number of good papers that have been submitted to ICLR this year, I'm recommending that this not be accepted at this time, but certainly hope that the authors continue to improve the paper towards a final publication at a different venue.